# Novelties and Perspectives of Intestinal Ultrasound in the Personalised Management of Patients with Inflammatory Bowel Diseases—A Systematic Review

**DOI:** 10.3390/diagnostics14080812

**Published:** 2024-04-12

**Authors:** Vasile-Claudiu Mihai, Liliana Gheorghe, Ioana-Irina Rezuș, Alina Ecaterina Jucan, Mihaela-Cristiana Andronic, Otilia Gavrilescu, Mihaela Dranga, Andrei-Mihai Andronic, Cristina Cijevschi Prelipcean, Ciprian Rezuș, Cătălina Mihai

**Affiliations:** 1Department of Radiology, University of Medicine and Pharmacy “Grigore T. Popa”, 700115 Iasi, Romania; 2Radiology Clinic, “St. Spiridon” County Clinical Emergency Hospital, Bulevardul Independentei 1, 700111 Iasi, Romania; 3Discipline of Gastroenterology, Medical Department I, University of Medicine and Pharmacy “Grigore T. Popa”, 700115 Iasi, Romania; ghiata.alina.ecaterina@gmail.com (A.E.J.); andronic_mihaela-cristiana@d.umfiasi.ro (M.-C.A.); otilia.gavrilescu@umfiasi.ro (O.G.); mihaela_dra@yahoo.com (M.D.); catalina.mihai@umfiasi.ro (C.M.); 4Institute of Gastroenterology and Hepatology, “St. Spiridon” County Clinical Emergency Hospital, Bulevardul Independentei 1, 700111 Iasi, Romania; cristina.cijevschi.prelipcean@umfiasi.ro; 5Discipline of Medical Semiology, Medical Department I, University of Medicine and Pharmacy “Grigore T. Popa”, 700115 Iasi, Romania; andrei.andronic@umfiasi.ro; 6Discipline of Internal Medicine, Medical Department I, University of Medicine and Pharmacy “Grigore T. Popa”, 700115 Iasi, Romania; ciprian.rezus@umfiasi.ro; 73rd Internal Medicine Clinic, “Sf. Spiridon” Emergency County Hospital, Bulevardul Independentei 1, 700111 Iasi, Romania

**Keywords:** Crohn’s disease, echography, disease follow-up, sonography, diagnostic

## Abstract

Inflammatory bowel diseases (IBDs) affect over 4.9 million individuals worldwide. Colonoscopy (CS) is the gold-standard technique for diagnosis. The remissive–recurrent pattern of evolution raises the need for non-invasive techniques to monitor disease activity. This review aims to present the advantages of intestinal ultrasound (IUS) in managing IBDs. Our search was conducted on the PubMed, Embase, and Cochrane (CENTRAL) databases, selecting original studies comparing IUS with other imaging and invasive monitoring methods. Our search yielded 8654 results, of which 107 met the inclusion criteria. Increased bowel wall thickness (BWT) and colour Doppler signal (CDS) are discriminative for disease activity. IUS can predict disease outcomes and detect response to treatment or postoperative recurrence. Contrast-enhanced ultrasound (CEUS) and elastography help differentiate fibrotic from inflammatory stenoses. The difficult rectal assessment limits the use of IUS in ulcerative colitis (UC). Transmural healing may develop as a therapeutic target as it is associated with better outcomes. Patients are compliant with this technique, and its results correlate well with CS and other imaging methods. In conclusion, IUS proves to be essential in assessing IBD activity and treatment response, predicting outcomes and detecting complications. CEUS and elastography are researched to improve the diagnostic values of IUS.

## 1. Introduction

Inflammatory bowel diseases (IBDs) are chronic immune-modulated diseases, mainly represented by Crohn’s disease (CD) and ulcerative colitis (UC), affecting over 10 million individuals globally [1]. Its pathogenesis is incompletely solved, incriminating genetic factors, intestinal microbiota, environmental factors, and immune system anomalies. Dysregulations of the innate immune system, through dendritic cells, macrophages, natural killer T cells, and tumour necrosis factor (TNF)-related cytokines, favour functional abnormalities of the adaptative immune system. Overexpression of T-helper 1 lymphocytes and stimulation of the interleukin-23/T-helper 17 pathway lead to sustained transmural inflammation in CD [2]. UC shares several common genetic loci and mechanisms with CD but is limited to the mucosa [3,4].

Both diseases are characterised by episodes of recurrence of different severities separated by periods of remission. The therapeutic management of IBDs remains a challenge, both regarding the selection of optimal treatment and the monitoring of the response to these medications.

Frequent re-evaluation is necessary to assess disease activity at different points in time. The current gold-standard technique for activity and extension exploration of IBDs is colonoscopy (CS), an invasive and costly technique with low repeatability. It is incompatible with frequent disease monitoring [5]. Non-invasive procedures include clinical parameters (different for CD and UC), blood (erythrocyte sedimentation rate—ESR; C-reactive protein—CRP), and stool (faecal calprotectin—FC) examinations. A plethora of biomarkers (such as genetic, metabolic, and bacterial flora) have been intensely studied, but they have yet to be put into use due to their contradictory results [6,7,8].

IUS is used frequently in IBD evaluation, as it is non-invasive, inexpensive, and highly tolerated by patients. Its capabilities for assessing transmural modifications, extraintestinal manifestations, and response to treatment are similar to those of other imaging and invasive methods.

The benefits of IUS in IBDs have been explored in a series of studies, with good-to-great results in CD, while colon assessment in patients with UC is still technically difficult [5]. The B-mode and Doppler US are readily available in most machines. They can offer a significant amount of information regarding bowel wall thickness (BWT), stratification (BWS), and vascularisation (assessed by colour Doppler signal—CDS) [9]. The mesenteric fat and lymph nodes can also be evaluated with these methods. The original and modified Limberg scores are frequently used to assess the CDS semi-quantitatively [10].

In addition to the classical B-mode and colour Doppler, a series of sonographic techniques may offer great value in managing patients with IBDs. Small intestine contrast ultrasonography (SICUS) represents an IUS examination conducted after oral ingestion of a contrast medium to distend the small bowel and enhance the visualisation of the intestinal walls [11]. Contrast-enhanced ultrasonography (CEUS) and elastography are more complex modes, requiring additional software for image acquisition and dedicated processing software for analysis. CEUS is an efficient tool for assessing disease activity based on bowel wall enhancement, complementary to Doppler examinations, and is highly accurate [12,13,14,15]. Elastography studies have gained popularity as a non-invasive exploration of tissue hardness, allowing for differentiation between fibrotic and inflammatory stenoses, both with different therapeutic management [16,17].

The IUS parameters listed previously have also been combined to create different scores to increase the sensitivity and specificity of diagnosis. A group of authors have developed the Milan Ultrasound Criteria (MUC = 1.4 × BWT (mm) + 2 × BWF, where BWF = 1 if present and BWF = 0 if absent, formerly known as the Humanitas Ultrasound Criteria) for UC and the Bowel Ultrasound Score (BUSS = 0.75 × BWT + 1.65 × BWF) for CD, with different cut-off values for differentiating between active and inactive disease [18,19,20,21]. The SUS-CD (Simple Ultrasound Score for CD) and IBUS-SAS (International Bowel Ultrasound Segmental Activity Score) have also been studied. Still, their introduction in clinical practice requires supplementary testing to determine their accuracy for disease evaluation.

Other imaging studies (Computed Tomography—CT; Magnetic Resonance Imaging—MRI) are essential in diagnosing, monitoring the evolution, and detecting complications associated with IBDs. Their use is especially important in CD, where cross-sectional explorations can evaluate transmural inflammation.

This article aims to analyse the current proof regarding the predictive value of classical IUS in patients with IBDs and the addition of complex sonographic techniques (CEUS, elastography) to enhance the accuracy of disease activity evaluation and personalised therapeutic management.

## 2. Materials and Methods

### 2.1. Literature Search and Search Terms

This systematic review was conducted following the PRISMA recommendations. A systematic search has been conducted on the PUBMED, Embase, and Cochrane databases from inception up to 31 October 2023. The search strategy contained the following terms: (“inflammatory bowel disease” OR “IBD” OR “Crohn’s disease” OR “Ulcerative colitis” OR “UC”) AND (“ultrasound” OR “ultrasonography” OR “US”) AND (“accuracy” OR “sensibility” OR “specificity” OR “positive predictive value” OR “negative predictive value” OR “score” OR “scoring” OR “index”), and no filter was applied.

### 2.2. Inclusions and Exclusion Criteria

The purpose of this search was to select studies related to the use of ultrasound in the follow-up of patients with IBDs regarding the response to treatment and outcome prediction. All types of original studies written in the English language from the last five years have been selected. Our eligibility criteria were original studies written in English from the last five years where bowel ultrasound (BUS) was used to measure disease activity and compared with Magnetic Resonance Enterography (MRE), Computed Tomography Enterography (CTE), capsule endoscopy (CE), double-balloon enteroscopy (DBE), or CS. Studies combining IUS and biochemical analysis (CRP, FC) were also included, as patients can benefit greatly from a multimodality disease evaluation. A manual search of the bibliographies of eligible studies was also performed. Studies in languages other than English and all articles comparing ultrasonography with scintigraphy or barium follow-through were excluded as these techniques are not applicable to the current diagnostic strategies. No review articles were accepted.

### 2.3. Study Selection and Data Extraction

We screened publications based on the title, abstract, and full text based on the inclusion and exclusion criteria. Two independent reviewers (V.-C.M. and I.-I.R.) analysed and selected studies respecting the inclusion and exclusion criteria. Any conflicts were solved by consensus. Of interest were articles showing the role of classical (B-mode, Doppler) and modern (CEUS, SICUS, elastography) ultrasound techniques in the follow-up of patients with IBDs. 

We created a standardised data collection sheet based on the consensus of methodological and clinical experts. We extracted the following data from the eligible articles: title, first author, year of publication, study design, main study findings, interventions, comparators, and outcomes (accuracy parameters, score formulas, evolution of disease parameters after treatment initiation, early predictors of lack of response to therapy and surgical risk, and predictors for disease evolution in the short and long term). One independent reviewer (V.-C.M.) extracted data using the standardised data collection form and double-checked the included information.

## 3. Results

Our search yielded 8654 studies. After duplicate removal, 5635 articles remained for screening based on the title and abstract. A total of 344 articles were selected for full-text analysis; of these, only 107 met the criteria for study inclusion (Figure 1).

Considering the differences between CD and UC, using IUS for different aspects of the two diseases will be treated separately in this article. The current treatment target for IBDs remains endoscopic healing, determined by the SES-CD (Simple Endoscopic Score for CD) and Mayo (UC) scores. Imaging studies are recommended as adjunct explorations for endoscopy, especially in CD, to assess a deeper level of healing by normalisation of IUS or MRE parameters [22]. Sonography has the advantages of being inexpensive, readily available in most clinics, non-radiating, and appropriate for multiple uses in a relatively short period.

BWT is measured in longitudinal and transversal planes, a mean of two measurements for each representing the accepted value. CDS is semi-quantitatively measured according to the Limberg score in five grades, from normal bowel wall with no vascular markings (grade 0) to long streaks of Doppler signal in the bowel wall extending to the mesentery (grade 4). It can also be dichotomised into two groups: normal (grades 0 and 1) and pathological CDS (≥grade 2) [23].

### 3.1. The Role of Ultrasound in Crohn’s Disease

#### 3.1.1. Disease Activity Evaluation

Disease activity evaluation in IBDs is required on multiple occasions, taking into consideration the remissive–recurrent nature of these diseases. There is no protocol regarding the frequency of imaging control. Still, patients with clinical remission and without sonographic evidence of inflammation demonstrated a lower risk of complications and treatment enhancement [7]. Therefore, a yearly IUS for patients with inactive disease and a 3- to 6-month period between evaluations for patients with active disease seems appropriate for follow-up [24]. CD’s most frequently affected bowel segment is the terminal ileum (TI), a region easily accessible by IUS [25,26,27]. However, the patchy pattern of intestinal involvement makes assessing the entire bowel, if possible, mandatory when evaluating these patients. 

Most studies have divided the analysed bowel into anatomical segments to determine the specific accuracy of each of these segments. An increased BWT (according to some authors ≥3 mm regardless of segment [7,24,28,29,30,31], while others consider ≥2 mm and ≥4 mm pathological for small intestine and colon disease, respectively [26,32,33,34]), loss of BWS (lack of differentiation between the five layers of the bowel wall), and positive CDS (ranging from small markings of Doppler signal to large streaks extending into the mesentery) are hallmarks of disease activity. Greater BWT and mucosal thickness were associated with greater histological scores [35]. Other pathological findings, including mesenteric fat echogenicity, have been associated with higher acute inflammatory scores [36,37,38]. Enlarged mesenteric lymph nodes and mesenteric fat hypertrophy are subjective and require more experience; thus, they are not included in the routine disease activity evaluation (Table 1).

The SUS-CD (mean AUC = 0.62) and IBUS-SAS (mean AUC = 0.55) showed various results regarding their accuracy in differentiating between active and inactive disease when compared to CS [44,49]. Quantitative values of CEUS performed well in differentiating between active and inactive endoscopic disease [33,41,49,50,51]. CEUS has been suggested as an alternative for the differentiation of active disease in patients with negative Doppler examination [34]. Oral contrast administration enhances the accuracy of IUS when compared to both MRE and CS. It is especially helpful in detecting structuring disease, almost in perfect agreement with CS [11].

#### 3.1.2. Prediction of Disease Evolution and Prognosis

Clinical remission is not necessarily associated with complete cessation of inflammation of bowel segments. In contrast, some symptomatic patients lack CS or imaging proof of disease activity [52]. These situations emphasise the need for patient-tailored evaluation, of which IUS shows great feasibility and good results in all aspects of activity evaluation and treatment monitoring. 

IUS can be used after initiating a new treatment or when increasing the dosage of the current therapeutic scheme to predict the response rate and to manage these patients efficiently. In patients with a continuous therapeutic scheme, IUS can be used to predict the risk for a negative disease course. This can be defined as the need for medication escalation, use of corticosteroids, hospitalisation due to symptomatic disease, or major surgery for complications associated with CD. Pathological BWT and the presence of BWF have been associated with negative disease evolution, while the presence of at least one complication at baseline IUS increases the risk of surgery [7,19,28,53]. 

Elastography studies have shown that patients with higher baseline strain ratio values may predict the need for CD-related surgical procedures. A strain ratio of ≥2 has been set as the cut-off for stratifying the patients at risk. This same study has found no significant differences in BWT, BWS pattern, or vascularisation in operated patients compared to those treated conservatively [29].

Serial IUS examinations can also stratify the patients requiring a therapeutic intervention from those responding well to the current treatment. Early normalisation of IUS parameters after treatment initiation is associated with better outcomes in the long term, highlighting the importance of sonographic evaluation in the prediction of disease evolution [26]. Clinical disease activity decline is associated with decreased BWT and progressive improvement of BWS. These outcomes are more evident in the first three months after adopting a novel therapeutic scheme, continuing until normalisation at a decreased rate in the following months [32].

Treatment modifications and supplementary examinations recommended based on IUS are superior to clinical decision making, leading to improved outcomes and better resource management (Table 2) [54,55].

#### 3.1.3. Complications Evaluation

The detection of certain complications can influence disease management. For example, dose escalation or corticosteroid administration to reduce patient symptoms can be detrimental in the presence of abscesses or fistulas, which require surgical and antibiotic treatment [55].

Stenoses, fistulas, and abscess formation are the most frequent CD-related complications. Bowel exploration by CS is limited in the presence of stenosis [30,31], while abscesses may be difficult to detect by this technique. IUS has shown promising results in complicated CD, while the addition of CEUS and elastography are used to enhance diagnostic accuracy (Table 3). Most articles have studied the TI and sigmoid colon only due to the difficulties of associating the localisation between different techniques.

There are three criteria required to assess a bowel segment as stenotic appropriately: increased BWT (>3 mm), luminal narrowing (<10 mm), and prestenotic dilatation (>30 mm). Visualising intestinal movement on IUS is another advantage of this technique, permitting the differentiation of normal peristalsis from stenosis. Fistulas can be defined as hypoechoic tracts disrupting the bowel wall, associated or not with the presence of gas or debris in their lumen [54,58,64].

Stenoses can be histologically classified as inflammatory or fibrotic. Their management differs, as the usual CD therapy can be efficient in predominantly inflammatory stenoses, while symptomatic fibrotic strictures require dilatation or surgical treatment. B-mode ultrasound is not particularly helpful in approximating the fibrosis grade. The absence of CDS is correlated to fibrotic stenoses, while increased CDS may indicate inflammatory aetiology [65,66]. Increased mean area under the curve (AUC) on CEUS is associated with higher fibrosis scores on surgical pieces [67]. One study failed to show a significant correlation between the number of vessels counted at histology and CDS or CEUS parameters [68]. Higher shear-wave elastography (SWE) values can differentiate severe (≥22.55 kPa) from mild and moderate (<22.55 kPa) fibrosis [65]. Higher values of SWE indicate an increased risk of surgery for CD-related complications, in particular, bowel obstruction due to stenoses [61,69]. More studies are required to determine the reliability and accuracy of these techniques.

Endoanal ultrasound (EAUS) with 3D reconstruction has also shown good precision in differentiating CD-associated fistulas from cryptoglandular fistulas based on four key aspects: a specific Crohn’s disease ultrasound fistula sign (CUFS), double-tract sign, maximum width, and the presence of debris in the fistulous tract [70]. Although less invasive than CS, this technique is also not highly acceptable. Transperineal ultrasound (TPUS) can accurately diagnose and follow up on the perianal fistulas and associated abscesses, with similar results compared to MRI. The localisation of the fistulous tract is described according to the anal clock face [60,62].

#### 3.1.4. Evaluation of Treatment Response

To date, transmural response (TR) has been defined as a reduction in BWT of at least 25% compared to baseline IUS or complete normalisation of BWT [8,27]. However, there has yet to be a consensus regarding the definition of TH, but some authors have stratified it into simplified, extended, and complete TH (Table 4). In addition, patients with TH may have a better evolution than those with mucosal healing alone, considering that the inflammatory process can still be active in the deeper layers of the bowel wall, which is not assessed by CS [26]. When possible, the length of disease involvement should also be studied; its decrease over time is a sign of responsiveness [63,71].

TI disease has been associated with a slower decrease in BWT compared to all the other bowel segments. Therefore, it is expected that the TR or TH in patients with this localisation are delayed [26,27,32]. Patients with previous use of biologics showed a delayed and less marked response to a change in the therapeutic scheme compared to biologic-naïve patients [27]. When considering a rate of decrease in the BWT of 0.004 mm per day, the need for accurate measurements is even more evident [71].

IUS evaluation is important to detect non-responders earlier and optimise their management to reduce the burden of IBDs. Some patients have demonstrated significant decreases in mean bowel wall length of involvement, BWT, and parietal and mesenteric CDS as early as two weeks after initiating a new therapeutic scheme [71,72]. Another study could define TR based on IUS four weeks after biologics initiation [27]. Successive sonographic explorations are useful in patients with response at the first follow-up IUS and those with minor or no response. In the first case, the improvement of bowel wall parameters is progressive, and re-evaluations are needed to show the evolution. At the same time, in the latter, a delayed response can be detected, preventing unnecessary treatment escalation. Early TH predicts TH at the follow-up evaluations and is associated with better outcomes, while many non-responders showed a negative disease evolution [73,74].

SWE has been introduced in clinical practice to determine tissue stiffness (fibrosis) non-invasively. Intravenous contrast can be administered to assess the degree of inflammation in the bowel wall accurately. Non-responders showed SWE values of ≥15.2 kPa at baseline [72]. A more severe inflammation, as determined by CEUS, is associated with a better response to therapy, possibly due to higher vascularisation, enhancing the medication infusion in the affected region [63].

CS can appreciate mucosal healing (MH) as the absence of inflammation (for example, healing of ulcers). Even though a few patients with TH show some degree of endoscopic activity, this case is more the exception than the rule, and this should not limit the use of IUS to evaluate therapeutic response (Table 5). In addition, the correlation between MH and TH is good to excellent in most studies [72]. Patients with TH require less frequent dose escalation or drug switches and need for corticosteroid use or hospitalisation [74].

#### 3.1.5. Postoperative Recurrence Detection

CD patients can undergo surgical procedures for complications (including abscesses, fistulas, obstructive symptoms, or penetration due to stenosis) or severe disease, non-responsive to medical treatment. Intestinal resection should be kept to a minimum in CD to prevent short bowel syndrome, but disease activity can appear in the remaining segments. Thus, re-evaluation is necessary to detect and treat the disease recurrence early. Based on its findings, a CS at 6–12 months postoperatively and ulterior personalised frequency of re-evaluation are used to detect recurrence [76,77]. IUS can ease the monitorisation of CD and has been studied in a few articles.

IUS correlated relatively well with CS findings [9], superior to clinical and inflammatory markers. When taking each abnormal parameter separately, loss of BWS and higher Limberg scores were more accurate than increased BWT [78]. In another study, bowel wall contrast enhancement (BWCE) ≥ 46% was accurate for disease activity, while the BWT and BWCE combined specificity was higher for active disease. BWT ≥ 6 mm, presence of complications, or BWT between 5 and 6 mm with BWCE ≥ 70% can be used to stratify patients with severe disease activity [79]. In patients with an endoanal plug for complex fistulas, EAUS proved to be useful in determining surgical failure and the need for reintervention [80].

### 3.2. The Role of Ultrasound in Ulcerative Colitis

UC manifests as continuous inflammation starting at the rectum and extending proximally, limited only to the colon. IUS has less widespread use in these patients, but they may benefit from non-invasive exploration in select conditions. The most frequent location of UC inflammation is the sigmoid colon, which is usually the most severely affected [14,26].

#### 3.2.1. Disease Activity Evaluation

BWT, CDS, and loss of BWS are also used in UC to assess disease activity. Of these, the BWT is the most strongly correlated with endoscopic findings [41,81,82,83]. Although 3 mm is considered the minimum value for an abnormal colon wall [54,84], statistical analysis in different studies has found different cut-offs with variate sensitivities and specificities (Table 6) [26,85,86]. The interrater agreement for the presence of disease activity was substantial, especially for BWT and CDS. In contrast, mesenteric fat hypertrophy, lymph nodes, loss of BWS, and loss of haustration showed only a fair agreement. The severity also depends on the IUS parameters, with higher disease activity exhibiting an increased risk of surgery [82,83,87].

Segmental accuracy is superior for IUS in the ascending and right colon, while CS can be used to assess the rectum [5,92]. The addition of sonographic evaluation to the CS and the introduction of an endoscopic ultrasound ulcerative colitis (EUS-UC) score can be used to determine the transmural inflammation even in UC and classify the patients according to severity [93]. 

The MUC, based on the BWT and CDS, is the most commonly used score to measure disease activity in UC patients [19,81]. Studies comparing IUS in healthy versus UC patients have also demonstrated increased BWT and elastography (ARFI—acoustic radiation force impulse) values in UC patients compared to controls [16]. SWE values are higher in active disease compared to remission, both when compared to the endoscopic index of severity and the clinical scores [17]. Although less used in UC, CEUS can enhance the diagnostic abilities of IUS, and the quantitative parameters PE and AUC have shown a good correlation with CS findings [41].

In UC, as opposed to CD, the clinical scores significantly correlate with disease activity as determined by IUS [6,94].

#### 3.2.2. Prediction of Disease Evolution and Prognosis

A negative disease course can be predicted based on the ultrasonographic disease activity. IUS-based decisions lead to better disease evolution, positive outcomes, and better resource management [56]. MUC has also been applied to stratify patients requiring treatment modifications or surgical procedures due to severe disease [18,95].

A particularity of UC compared to CD is that acute severe disease can have a major impact on patient outcomes, with surgical excision remaining the last therapeutic solution. A BWT > 3.4 mm, increased vascularity, and loss of BWS are significant predictors of patients not responding to initial corticosteroid therapy [86]. Dynamic values can also be used to predict non-responsiveness [96]. Salvage therapy with Infliximab, when applied under frequent IUS exploration, permits the determination of the response or the need for surgery. BWT is significantly lower, and it follows a decreasing pattern in steroid-responsive patients, while significant differences in responders to salvage therapy from non-responders could not be found [97]. 

Higher EUS-UC scores are associated with therapeutic escalation and colectomy in the short (3 months) and long term (2 years) [98].

#### 3.2.3. Evaluation of Treatment Response

IUS has successfully identified patients requiring modification of the therapeutic scheme, and early changes lead to better outcomes (Table 7) [99]. In acute settings, increased BWT (>3.4 mm) and loss of BWS are associated with steroid resistance [88]. Although current guidelines do not consider these parameters for treatment decisions, further studies may prove IUS-based decisions to show superior response and prevent surgery. The EUS-UC score has also been used to evaluate response to therapy; its decrease over time leads to bowel normalisation and better outcomes [93].

## 4. Discussion

IUS is an examination that can be repeated multiple times at different presentations to assess the status of the patient’s disease. Although CS is the gold standard technique for diagnosis, its use in asymptomatic patients is limited, with IUS being a great tool for this role. Most patients find it highly acceptable and useful, causing little to no discomfort and permitting the return to normal activities on the same day [100,101]. IUS enhances the patient’s knowledge, leading to a better understanding of the disease’s overall symptoms, activity, and management, as well as the need to comply with treatment recommendations, regardless of level of education or disease type. When asked about their preference, IUS was ranked highest by patients [101,102,103]. 

Of the main pathological findings, BWT and CDS are suggested to be the most reliable for disease assessment. A few attempts have been made to enhance the utility of BWS, mesenteric fat hypertrophy and lymph node enlargement, but they are less frequent and more subjective [104,105]. A plethora of ultrasonographic scores obtained by combining these parameters have been imagined by numerous authors and showed various accuracies for disease activity evaluation [106,107]. However, there are currently no meta-analyses or systematic reviews comparing them to determine which ones are the most appropriate for clinical settings.

The goals of IBD response to treatment are evolving towards a deeper degree of healing that can be assessed only by cross-sectional imaging methods. Transmural healing is associated with reduced hospitalisation rates for recurrent disease or complication, decreased need for surgery, and long-term corticosteroid-free remission [106,108].

CEUS, SICUS, and elastography are additional tools that may be complementary to IUS in disease evaluation. Although they participate in prolonging the examination times, they tend to be more accurate in determining inflammatory activity and the presence of complications [106].

IUS has a few limitations, including the difficulty in viewing rectal disease or bowel segments hidden behind gas, but multiple studies have shown its accuracy in evaluating disease activity and detecting complications. Transperineal and endoanal US may be useful in assessing rectal and perianal disease [60,62,64].

In CD patients, negative ileo-colonoscopy is not necessarily indicative of disease inactivity. Clinical activity scores after treatment initiation are relevant to patients, but up to 50% of individuals can exhibit endoscopic or imaging proof of inflammation [84]. Both CE and IUS can be used to assess small intestinal disease, but there is insufficient evidence regarding the correlation levels and accuracy [109].

Special groups of patients are represented by pregnant women and children, in whom irradiating methods (CTE) are contraindicated as they pose great risks for the foetus and the underage population [110]. In both groups, MRE is difficult to obtain, either due to gantry dimensions in female patients with voluminous abdomen or due to lack of cooperation in paediatric patients [107].

Pregnancy can alter the clinical presentation of IBDs, while some of the symptoms can be related to their status. Some of the biochemical parameters can be modified in pregnant females and are unable to offer information about the extent, location, or complications [99]. These needs are met by IUS, which, in addition to the activity status and extraintestinal manifestations, is able to diagnose concurrent conditions. It showed overall high sensitivity, specificity, and negative predictive value for disease evaluation. The same standardised evaluation protocol can be used in pregnant patients. However, during late pregnancy, the feasibility of IUS decreases due to the voluminous uterus, especially in viewing the TI and ileocecal valve, with adequate views obtained in only 50% of scans. Significant differences in the size of the uterus have decreased IUS’s usefulness [99,106,110,111].

Paediatric patients benefit greatly from IUS. Extensive disease evolution is more frequent in children compared to adults, especially in the small bowel, and this poses a greater threat to their development [52,84]. The pathological findings are similar to those of adults, with BWT remaining the most reliable predictor of disease. The particularities of the inflammatory reaction increase the relevance of altered bowel pattern, mesenteric fat hypertrophy and lymph node enlargement in children. However, these findings are highly non-specific, and their presence alone should not be considered an absolute diagnostic marker [37,40]. Radiological remission in this group has also been suggested to greatly reduce the risk of recurrence or complications [52,105]. Contrast administration in children has been approved by the United States Food and Drug Administration in 2016, but its use in this population remains to be determined [67]. 

MRE is considered the most accurate imaging method for disease evaluation. Still, its use is limited by the low availability, long duration for image acquisition, and high costs, making it less likely to be frequently used to examine these patients. Technical difficulties such as suboptimal distension, inability to tolerate oral contrast medium, or motion artefacts are detrimental to accurate assessment and are more frequent with MRE than IUS. MRE is considered superior to IUS for determining disease extension and detection of complications [25,112]. It can be conducted in various conditions, being unaffected by the gaseous contents of the bowel [113]. A series of studies have demonstrated divergent results regarding the correlation between MRE and IUS regarding disease activity assessment [30,114,115,116].

CT is great in emergency settings and has relatively good sensitivity for disease complications. Its findings are also relatively non-specific according to the current studies, with CEUS showing superiority [117]. The main concern is related to patient exposure to ionising radiation. 

Our study has several strengths and limitations. First of all, we followed a strict methodology based on the PRISMA recommendations and performed a comprehensive analysis of the literature. Our comparison was extensive as IUS was faced with each of the clinically available tools (MRE, CTE, CS) used currently to assess IBDs. Another strength is that we also included evidence on special populations (paediatric patients and during pregnancy).

The limitations of our study come from the great heterogeneity of the definitions of abnormal BWT or BWS patterns. For instance, for TI, both 2 and 3 mm have been used as cut-offs, while 3 or 4 mm values have been considered for colonic disease [24,26,28,29,30,31,32,33,34,99]. For CDS, some studies have attempted to semi-quantitatively approximate the degree of hypervascularisation by using the Limberg score, while other authors preferred a dichotomous approach, with present or absent CDS [10,23,32,48,78]. Considering the many available abnormal findings, the decision on which parameters to follow is difficult and requires unification. Secondly, the time between the analysed procedures was different in every study, ranging from one day [37] to thirty days [88], limiting the comparative value as disease activity can be modified by time. Therefore, standardised acquisition and reporting protocols are the long-term goals of IUS.

One study reported significantly higher values of mucosal thickness in UC patients, while the submucosal thickness was higher in those with CD [112]. In the future, studies evaluating such differences can accurately differentiate between the two conditions, limiting the number of invasive procedures to uncertain cases. Artificial intelligence (AI) may greatly aid the development of various criteria, simplifying diagnostic procedures.

## 5. Conclusions

In conclusion, IUS is an inexpensive, readily available, non-invasive cross-sectional imaging method that greatly benefits the management of IBDs and opens a path to a personalised approach for these patients. Future perspectives include improving this exploration by using CEUS and elastography, while AI may enhance diagnostic possibilities. Supplementary studies, and especially randomised control trials, may be useful in filling the gap in the knowledge regarding the use of IUS for patients with IBDs and in standardising the method.

## Figures and Tables

**Figure 1 diagnostics-14-00812-f001:**
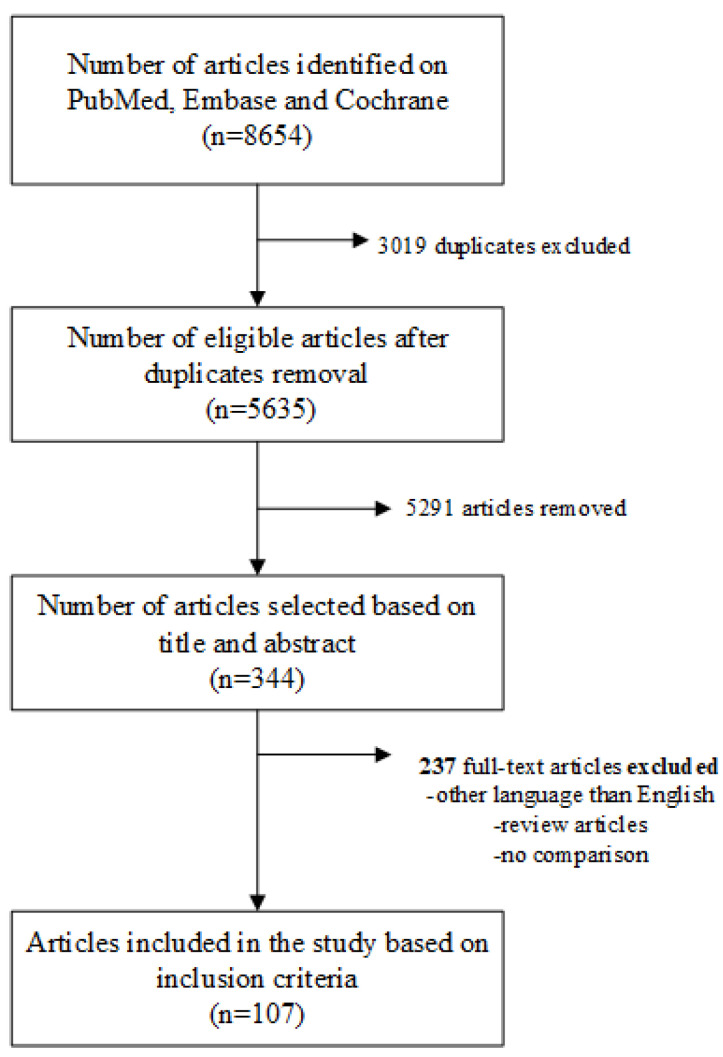
PRISMA flowchart for the study selection process and results.

**Table 1 diagnostics-14-00812-t001:** Articles showcasing the role of IUS in the evaluation of disease activity.

Study	Study Design	No. of Patients	Age	Significant Parameters	Accuracy	Comparator
Calavrezos et al. [39]	Retrospective	44	29.5 (21–47)	BWT > 3 mm increased CDS	Se 88.1%, Sp 50%, PPV = 86.1, NPV = 54.6, Accuracy 79.6%	CS, MRE
Dell’Era et al. [40]	Retrospective	113	10.8 ± 4.0	Increased BWT (>3 mm)	Se 69.6%, Sp 96.7%, PPV = 84.2%, NPV = 92.6	CS
				Altered bowel pattern	Se 78.3%, Sp 93.3%, PPV = 75%, NPV = 94.4%	
				i-fat	Se 65.2%, Sp 92.2%, PPV = 68.2%, NPV = 91.2%	
Ma et al. [41]	Prospective	15	36.87 ± 16.14	BWT > 4.7 mm	Se 90%, Sp 80%	CS, CT
				PE (44.37 dB)	Se 75%, Sp 100%	
				TTP (10.73 s)	Se 94%, Sp 25%	
				AUC (226.15 dBsec)	Se 81%, Sp 75%	
Novak et al. [42]	Retrospective and prospective	160 + 87	34.9 (25–48)	SSS (based on BWT and CDS)	Se 92.1–93.3%, Sp 76.8–81.6%, PPV = 79.6–88.2%, NPV = 86.3–93.2%, Accuracy 86.2–87.5%	CS
Ponorac et al. [13]	Prospective	24	14 (13–17)	BWT, CDS, loss of BWS, i-fat	Se 55.6%, Sp 86.4%, Accuracy 72.5%	Histopathological result
Ripolles et al. [33]	Prospective	72	45.6	Simple CEUS score = (BWT × 0.957) + (CDS × 0.859) + (wash-in × 0.036).	Se 94%, Sp 91%, PPV 95.9%, NPV 87%	CS
Saevik et al. [43]	Prospective	164	41.5 (18–83)	SUS-CD = BWT × 1.053 + CDS × 1.934 + i-fat × 1.275 + BWS × 1.225 + 0.242	Se 95.3%, Sp 70.3%, AUC 0.92	CS
Saevik et al. [44]	Prospective	145	18–83	BWT ≥ 3 mm	Se 92.2%, Sp 86%, PPV 94%, NPV 82.2%	CS
				BWT ≥ 4 mm	Se 80.4%, Sp 90.7%, PPV 95.3%, NPV 66.1%	
				CDS	Se 66.7%, Sp 97.7, PPV 98.6%, NPV 55.3%	
Smith et al. [45]	Retrospective	8	54 (35–65)	BWT 5.0 mm (4.4–6.7) increased CDS	N/A	CS
Tsai et al. [46]	Prospective	41	13.7 (4.6–18.9)	Ileal BWT > 1.9 (1.8) mm	Se > 91%	CS, MRE
Wright et al. [47]	Retrospective	65	-	BWT > 3 mm ± increased CDS, loss of BWS, i-fat, LN	Se 72%, Sp 86.7%	CS
					Se 87.5%, Sp 61.1%	MRE
Xu et al. [48]	Retrospective	115	31 (24–42)	BWT > 3 mm, loss of BWS, i-fat, CDS (Limberg score)	Se 90.9% (91.9% by adding CDS)	CS, MRE, CTE

AUC = area under the curve; BWT = bowel wall thickness; CDS = colour Doppler signal; CEUS = contrast-enhanced ultrasound; CS = colonoscopy; CTE = Computed Tomography Enterography; i-fat = mesenteric fat hypertrophy; LN = lymph node enlargement; MRE = Magnetic Resonance Enterography; NPV = negative predictive value; PPV = positive predictive value; Se = sensitivity; SSS = Simple Sonographic Score, calculated with the formula (0.0563 × BWT1) + (2.0047 × BWT2) + (3.0881 × BWT3) + (1.0204 × CDS1) + (1.5460 × CDS2); Sp = specificity; SUS-CD = Simple Ultrasound Score in Crohn’s Disease.

**Table 2 diagnostics-14-00812-t002:** Articles studying the role of IUS in disease evolution prediction.

Study	Study Design	No. of Patients	Frequency of Re-Evaluation	IUS Parameters	Outcome with Parameter
Alloca et al. [19]	Prospective	225	Baseline, 12 months	BUSS ≥ 3.52 (0.75 × BWT + 1.65 × BWF)	Overall negative course, treatment escalation
				Complications at baseline	Overall negative course, surgery
Helwig et al. [26]	Prospective	234	Baseline, 12 and 52 weeks	Patients with TR or simplified TH at 12 weeks (see text) had a better evolution at 52 weeks	
Kucharzik et al. [32]	Prospective	234	Baseline, then at 3, 6, and 12 months	BWT, BWS, CDS (Limberg)	Clinical and biochemical improvement
Les et al. [56]	Prospective	89	Baseline, 6 months	BUS score > 0.45 (=1/(1 + EXP(−0.88 × BWT + 2.02 × Doppler − 6.67)	Need for IBD treatment intensification in 6 months
Les et al. [56]	Prospective	89	Baseline, 6 months	BUS score > 0.5 (=1/(1 + EXP (−0.75 × BWT + 3.5 × Doppler − 7.31)	Immediate treatment intensification due to active disease
				BUS score > 0.6 (=1/(1 + EXP (−0.8 × BWT − 1.3 × BWS − 3.82)	Subsequent treatment intensification in a 6-month time frame
Rispo et al. [28]	Prospective	100	-	BWT ≥ 7 mm, SB extension ≥ 33 cm, stricturing/penetrating disease	Need for surgery in a 1-year time frame
Rueda Garcia et al. [53]	Retrospective	70	-	Pathological CDS, presence of fistulas or abscesses	Increased risk of surgery
Vaughan et al. [7]	Retrospective	202	-	Sonographic inflammation: abnormal BWT, CDS, BWS, or i-fat	Univariate analysis: use of corticosteroids, reduced hospitalisation-free and surgery-free survival
					Multivariate analysis: use of corticosteroids
Quaia et al. [57]	Prospective	115	Baseline and at ≈6 weeks	Higher values of pretreatment PE, AUC wash-in, AUC wash-out	More severe inflammation is associated with a better response to treatment

BUS = bowel ultrasound; BUSS = Bowel Ultrasound Score; BWF = bowel wall flow; BWS = bowel wall stratification; BWT = bowel wall thickness; CDS = colour Doppler signal; i-fat = mesenteric fat hypertrophy; PE = peak enhancement; SB = small bowel; TH = transmural healing; TR = transmural response.

**Table 3 diagnostics-14-00812-t003:** Accuracy of IUS in the diagnosis of complicated CD.

Study	Study Design	No. of Patients	Complication by IUS	Se (%)	Sp (%)	Acc (%)	PPV	NPV
Alloca et al. [58]	Prospective	17	Stenosis	93 (68–100)	86 (42–100)	91 (71–99)	93 (68–100)	86 (42–100)
Calavrezos et al. [39]	Retrospective	44	Stenosis	57.1	91.5	87	50	93.5
			Fistula	57.1	100	94.4	100	95
			Abscess	60	95.5	88.9	75	91.3
Ding et al. [59]	Prospective	25	p-SWE for differentiating inflammatory from fibrotic stenoses	75	100	96	100	95.5
Hakim et al. [11]	Retrospective	93	Stenosis (by SICUS)	67	100	94.1	100	93.3
Jung et al. [60]	Retrospective	29	Perianal fistula (TPUS, evolution under treatment)	63.3	93.3	73.3	95	56
Kamel et al. [61]	Prospective	26	Fistula	85.7	100	95	100	92.9
			Stricture	100	94.4	95	66.7	100
			Abscess	100	100	100	100	100
Lee et al. [62]	Retrospective	38	Perianal fistula (TPUS)	76.3	53.3	-	84.2	40.8
			Perianal abscess (TPUS)	56.3	98.1	-	90	88
Quaia et al. [63]	Prospective	20	Stenosis detected by B-mode + CEUS + SE	35/45	25/40	70/75	-	-
Wright et al. [47]	Retrospective	65	Strictures, fistulas, abscesses	85.7	94.3	-	-	-

Acc = accuracy; CEUS = contrast-enhanced ultrasonography; NPV = negative predictive value; p-SWE = point shear-wave elastography; PPV = positive predictive value; SE = strain elastography; SICUS = small intestine contrast ultrasonography; Se = sensitivity; Sp = specificity; TPUS = transperineal ultrasonography.

**Table 4 diagnostics-14-00812-t004:** Definitions of TR and TH, after Helwig et al. [26].

Type of Response	BWT Normalisation	CDS Normalisation	Normal BWS	No i-fat
**TR**	X			
Simplified TH	X	X		
Extended TH	X	X	X	X
**Two out of three parameters assessed; one could not be assessed/documented**				
Complete TH	X	X	X	X
**All three parameters assessed**				

BWS = bowel wall stratification; BWT = bowel wall thickness; CDS = colour Doppler signal; i-fat = mesenteric fat hypertrophy; TH = transmural healing; TR = transmural response.

**Table 5 diagnostics-14-00812-t005:** Articles studying the IUS outcomes of patients started on a new therapeutic scheme.

Study	Study Design	No. of Patients	Treatment	IUS Frequency of Re-Evaluation	IUS Outcome	Additional Outcomes
Kucharzik et al. [27]	Interventional (randomised)	77	Ustekinumab	Baseline, then at 4, 8, 16, and 32 weeks	TR (reduction of BWT ≥ 25% or normalisation of all IUS parameters)	Progressive improvement of BWT, CDS, BWS, i-fat
Dillman et al. [71]	Prospective observational	28	Infliximab ± Azathioprine	Baseline, then at ≈2 weeks and ≈1, 3, and 6 months	Reduction of length of bowel wall disease involvement, BWT, CDS	-
Hoffman et al. [23]	Interventional (non-randomised)	23	Ustekinumab	Before the first (week 0) and second (week 8) administration	Reduction of BWT of at least 1 mm	Decrease in CDAI (at least 70), CRP (≥0.5 mg/dL)
Orlando et al. [29]	Interventional (non-randomised)	30	Infliximab or Adalimumab	Baseline, then at 14 and 52 weeks	Reduction of BWT (normal BWT considered TH)	SR ≥ 2 indicates non-responders and increased risk of surgery
Paredes et al. [73]	Interventional (non-randomised)	36	Infliximab or Adalimumab	Baseline, then at 12 weeks and 1 year	TH (BWT ≤ 3 mm, colour Doppler grade 0 or 1), improvement (≥2 mm decrease in BWT and ≥1 grade of CDS)	Partial (100 points CDAI decrease) or complete (150 points) clinical response, clinical remission
Zorzi et al. [75]	Prospective observational	80	Infliximab or Adalimumab	Baseline and 18 (12–24) months	Improvement (≥1 mm) or normalisation of BWT, decreased length of disease, TH associated with better outcomes	-
Castiglione et al. [74]	Prospective observational	218	Infliximab or Adalimumab	Every 3 months for at least 1 year	TH is associated with reduced rates of hospitalisation, surgery or need for corticosteroids	MH is also associated with better outcomes than no healing group
Chen et al. [72]	Prospective observational	30	Infliximab	Baseline, then at weeks 2, 6, and 14	Decrease in BWT or TH (BWT ≤ 3 mm), SWE as a predictor of non-responsiveness (cut-off 15.2 kPa)	Clinical response (ΔCDAI ≥ −100) or remission (CDAI < 150), mucosal healing
Quaia et al. [63]	Prospective observational	115	Infliximab or Adalimumab	Baseline and at ≈6 weeks	Decrease in BWT in responders; higher PE, WIAUC and WOAUC in responders at baseline vs. post-treatment	-

BWS = bowel wall stratification; BWT = bowel wall thickness; CDAI = Crohn’s Disease Activity Index; CDS = colour Doppler signal; CRP = C-reactive protein; i-fat = mesenteric fat hypertrophy; IUS = intestinal ultrasound; PE = peak enhancement; SWE = shear-wave elastography; TH = transmural healing; TR = transmural response; WIAUC = wash-in area under the curve; WOAUC = wash-out area under the curve.

**Table 6 diagnostics-14-00812-t006:** Articles studying the accuracy of IUS for UC disease activity.

Study	Study Design	No. of Patients	Active Disease Parameter(s)	Se (%)	Sp (%)	PPV (%)	NPV (%)	AUC	Acc (%)	Comparator
Alloca et al. [19]	Prospective	43	MUC (=1.4 × BWT + 2 × CDS) > 6.2 (development phase)	71	100	-	-	0.891	-	Mayo Endoscopic Score
			MUC (=1.4 × BWT + 2 × CDS) > 6.2 (validation phase)	85	94	-	-	0.902	-	
Sathananthan et al. [88]	Prospective	39	BWT > 3 mm + increased CDS (POCUS)	92	86	92	86	-	-	CS within 30 days
				100	100	100	100			CS on the same day
			Disease extent	92	80	80	86	-	-	CS
Myioshi et al. [89]	Retrospective	29	BWT ≥ 3.75	73.3	93.2	88	83.7	-	-	CS
			LWS	49.1	84.2	90	36.4	-	-	
			SMI (=SMT/BWT × 100) ≥ 49.7	83.3	81.8	75.8	87.8	-	-	
			Positive CDS	76.7	90.9	85.2	85.1	-	-	
Lim et al. [90] Lim et al.	Prospective Prospective	29 29	BWT > 3 mm, LWS, increased CDS, i-fat	50	100	100	84	-	-	CS
								-	-	
			BWT > 3 mm	33.3	65.4	10	89.5	-	-	
			LWS	33.3	84.6	20	91.7	-	-	
			i-fat	33.3	92.3	33.3	92.3	-	-	
			Increased CDS	33.3	92.3	33.3	92.3	-	-	
Goodsall et al. [81]	Prospective	19	MUC > 6.3	55	100	100	31	-	-	Histology
			MUC > 6.3 and FC > 100 μg/g	88	80	95	57	-	-	
Stojkovic et al. [82]	Retrospective	55	BWT, CDS, i-fat, LWS, LN	87.8	83.3	-	-	-	-	CS
			BWT > 4.75 mm for moderate–severe disease	82	64	-	-	-	-	
Takahara et al. [85]	Prospective	80	BWT > 2 mm, MES > 0	94	77	93	80		90	CS
van Wassenaer et al. [91]	Prospective	35	UC-IUS > 1 for MES ≥ 2	88–100	83–87			0.82–0.88		CS
			Civitelly index > 2	6–25	93–100			0.76–0.84		
Yamada et al. [17]	Prospective	26	SWE > 2.2 m/s	86.4	75			0.909		CS

Acc = accuracy; AUC = area under the curve; BWT = bowel wall thickness; CDS = colour Doppler signal; CS = colonoscopy; FC = faecal calprotectin; i-fat = mesenteric fat hypertrophy; MES = Mayo Endoscopic Score; MUC = Milan Ultrasound Criteria; NPV = negative predicting value; PPV = positive predictive value; Se = sensitivity; SMI = submucosal index; SMT = submucosal thickness; Sp = specificity; SWE = shear-wave elastography.

**Table 7 diagnostics-14-00812-t007:** Articles exploring the use of IUS to follow up patients after treatment initiation in UC.

Study	Study Design	No. of Patients	Treatment	IUS Frequency of Re-Evaluation	IUS Outcome
Jin et al. [93]	Prospective	79	Mesalazine ± steroids, Infliximab	Baseline, then at 2 and 6 months	Decrease in EUS-UC score at the follow-up visits
Smith et al. [97]	Prospective	10	Steroids and Infliximab for acute severe UC	First 24 h, then at 3 and 7 days	Lower BWT and significant decrease in responders
Helwig et al. [26]	Prospective	171	Corticosteroids, AZA/6-MP, anti-TNF, anti-integrin	Baseline and 12 weeks	TR and TH at 12 weeks are associated with better outcomes
Goertz et al. [14]	Prospective	7	Vedolizumab	Baseline and 14 weeks	A slight decrease in BWT and CDS grade in responders, increase in non-responders a significant decrease in WiR in responders
Ilvemark et al. [96]	Prospective	56	Corticosteroids for acute severe UC	Before treatment, at 24 ± 24 h and 6 ± 1 days	BWT ≥ 4 mm and BWT reduction ≤ 1 mm or ≤20% in non-responders at 48 h

6-MP = 6 mercaptopurine; anti-TNF = anti-tumour necrosis factor; AZA = azathioprine; BWT = bowel wall thickness; CDS = colour Doppler signal; EUS-UC = endoscopic ultrasonography ulcerative colitis score; UC = ulcerative colitis; WiR = wash-in rate.

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
