# Peer review of "Novelties and Perspectives of Intestinal Ultrasound in the Personalised Management of Patients with Inflammatory Bowel Diseases—A Systematic Review"

_diagnostics, 2024, doi:10.3390/diagnostics14080812_

Round 1

Reviewer 1 Report

Comments and Suggestions for Authors

A pertinent review. 

The following are suggested to improve the article:

1. The limitations of the study have not been mentioned 

2. The discussion needs to be improved. The following may be discussed and referenced:

a. Lin WC, Chang CW, Chen MJ, Wang HY. Intestinal Ultrasound in Inflammatory Bowel Disease: A Novel and Increasingly Important Tool. J Med Ultrasound. 2023 Jan 19;31(2):86-91. doi: 10.4103/jmu.jmu_84_22. PMID: 37576427; PMCID: PMC10413392.

b. Nardone OM, Calabrese G, Testa A, Caiazzo A, Fierro G, Rispo A, Castiglione F. The Impact of Intestinal Ultrasound on the Management of Inflammatory Bowel Disease: From Established Facts Toward New Horizons. Front Med (Lausanne). 2022 May 23;9:898092. doi: 10.3389/fmed.2022.898092. PMID: 35677820; PMCID: PMC9167952.

c. Dolinger MT, Calabrese E, Pizzolante F, Abreu MT. Current and Novel Uses of Intestinal Ultrasound in Inflammatory Bowel Disease. Gastroenterol Hepatol (N Y). 2023 Aug;19(8):447-457. PMID: 37772159; PMCID: PMC10524432.

d. Bots S, De Voogd F, De Jong M, Ligtvoet V, Löwenberg M, Duijvestein M, Ponsioen CY, D'Haens G, Gecse KB. Point-of-care Intestinal Ultrasound in IBD Patients: Disease Management and Diagnostic Yield in a Real-world Cohort and Proposal of a Point-of-care Algorithm. J Crohns Colitis. 2022 May 10;16(4):606-615. doi: 10.1093/ecco-jcc/jjab175. PMID: 34636839; PMCID: PMC9089417.

e. Kellar A, Dolinger M, Novak KL, Chavannes M, Dubinsky M, Huynh H. Intestinal Ultrasound for the Pediatric Gastroenterologist: A Guide for Inflammatory Bowel Disease Monitoring in Children: Expert Consensus on Behalf of the International Bowel Ultrasound Group (IBUS) Pediatric Committee. J Pediatr Gastroenterol Nutr. 2023 Feb 1;76(2):142-148. doi: 10.1097/MPG.0000000000003649. Epub 2022 Oct 28. PMID: 36306530; PMCID: PMC9848217.

f. Dolinger MT, Kayal M. Intestinal ultrasound as a non-invasive tool to monitor inflammatory bowel disease activity and guide clinical decision making. World J Gastroenterol. 2023 Apr 21;29(15):2272-2282. doi: 10.3748/wjg.v29.i15.2272. PMID: 37124889; PMCID: PMC10134421.

g. Rajagopalan A, Sathananthan D, An YK, Van De Ven L, Martin S, Fon J, Costello SP, Begun J, Bryant RV. Gastrointestinal ultrasound in inflammatory bowel disease care: Patient perceptions and impact on disease-related knowledge. JGH Open. 2019 Oct 9;4(2):267-272. doi: 10.1002/jgh3.12268. PMID: 32280776; PMCID: PMC7144798.

Comments on the Quality of English Language

The language is largely correct

Author Response

Dear reviewer,

Thank you for your suggestions, I found them highly valuable.

I am attaching the corrected manuscript with all modifications highlighted. I have added a paragraph for limitations (lines 530-540) and further expanded the discussions section, giving more details about the uses of IUS in IBD (lines 456-472) and particular group populations (lines 482-509). I also made some small modifications to the English language in the text.

Thank you for your kind help!

Please let me know if you find other modifications that may require my attention.

Reviewer 2 Report

Comments and Suggestions for Authors

The authors shed light on the value of IUS in IBD diagnosis, management, and prognosis. IUS is a new non-invasive modality that can be used in chronic diseases that are characterized by bouts of remission and exacerbations. This is the case in IBD. The main value of IUS is in Crohn's disease with limited value in Ulcerative Colitis. Undoubtedly, it is too early to judge the value of IUS in disease diagnosis and follow-up, however, the authors made a huge effort in collecting their data and comparing IUS with other used diagnostic modalities.

Few comments:

Abstract needs to be rephrased with the removal of the methodology and the addition of the conclusions.

You have to add before the introduction a section for abbreviations.

If you can make the introduction more concise.

This is a review article and you do not need to be strict with the introduction, material and method, and result form. Better to admix both the material and methods sections with the result section.

The section on the role of IUS in UC diagnosis better be more concise.

Discussion:

The third paragraph from line 436 to line 455 is better to come first in the discussion followed by the first and second paragraphs from line 422 to line 435 i.e. IUS is an examination that can be repeated multiple times at different presentations .....then, MRE is considered the most accurate imaging method for.... 

The last 2 paragraphs from line 456 to line 466 had to be admixed together as study limitations.

Author Response

Dear reviewer,

Thank you for your suggestions. I found them highly valuable.

I am sending the corrected manuscripts, with all the modifications highlighted.

I changed the abstract by adding the conclusions of my study (lines 39-42), but I also kept a short phrase about the methods (according to PRISMA abstract guidelines).

I introduced a section for abbreviations (lines 48-68).

We rephrased the introduction and changed the order of the paragraphs so that IUS comes first. 

We followed the PRISMA recommendations regarding the structure of the article and thought methods should remain from results.

Regarding the use of IUS in UC, we believe that it is important to highlight the current status of ultrasonography in this condition, as patients can benefit similarly to those with CD. The UC section is already shorter, as there is fewer data on the subject, but we hope that our article will highlight the gaps in the knowledge and be used as a starting point for future studies.

I changed the order of paragraphs in discussions so that IUS (lines 447-509) is presented before other imaging techniques (lines 510-523) and added the study limitations. 

Thank you for your help!

Please let me know if you find other modifications that will require my attention.

Reviewer 3 Report

Comments and Suggestions for Authors

The Introduction can focus more on the benefits of intestinal ultrasound (IUS). Also, IUS can be listed earlier than CT.

Tables, when applicable, should have a header if they are continued on the next page so the readers will understand the content easier.

In the Discussion, a few additional sentences can be included to describe the benefits of IUS in more detail for the pediatric and pregnant patient populations.

The Discussion should focus more on IUS and start out with its details.

Thank you.

Comments on the Quality of English Language

English is good.

Author Response

Dear reviewer,

Thank you for your suggestions. I found them highly valuable.

I am attaching the corrected manuscript with all the modifications highlighted.

I have rephrased the introduction, adding some information about the advantages of IUS and changed the order of the paragraphs so that it comes first, before the other imaging methods (lines 92-128).

I added headers to all tables continuing on more than 1 page.

I have added a few paragraphs about special populations (lines 482-509) and also further expanded the paragraphs about the importance of IUS in IBD (which now come first, before other imaging methods, respectively in lines 447-481) in the discussions section.

I also made small corrections in the English language used in the article.

Thank you for your help!

Please let me know if there are more modifications that I may need to make!

Round 2

Reviewer 1 Report

Comments and Suggestions for Authors

All issues addressed as the article has been revised as per suggestions

The discussion reads better now and all aspects have been covered